# Compact Wideband Double-Slot Microstrip Feed Engraved TEM Horn Strip Antennas on a Multilayer Substrate Board for in Bed Resting Body Positions Determination Based on Artificial Intelligence

**DOI:** 10.3390/s22239555

**Published:** 2022-12-06

**Authors:** Jiwan Ghimire, Ji-Hoon Kim, Dong-You Choi

**Affiliations:** 1Department of Information and Communication Engineering, Chosun University, Gwangju 61452, Republic of Korea; 2Department of Mechanical Engineering, Chosun University, Gwangju 61452, Republic of Korea

**Keywords:** UBW, TEM horn, microstrip power divider, radar sensors, Recurrent Neural Network, artificial intelligence

## Abstract

In this paper, a horn-shaped strip antenna exponentially tapered carved on a multilayer dielectric substrate for an indoor body position tracking system is proposed. The performance of the proposed antenna was verified by testing it as a tracking state of an indoor resting body position. Among different feeding techniques, the uniplanar T-junction power divider approach is used. The performance verification of the proposed antenna is explained through its compact size and 3D shape, along with a performance comparison of the return loss radiation pattern and the realized gain. The suggested antenna has an 88.88% fractional bandwidth and a return loss between 6 and 15.6 GHz, with a maximum gain of 9.46 dBi in the 9.5 GHz region. Within the intended band, the radiation pattern had an excellent directivity characteristics. The proposed antenna was connected to an NVA-R661 module of Xethru Inc. for sleeping body position tracking. The performance of the antenna is measured through microwave imagining of the state of the resting body in various sleeping positions on the bed using a Recurrent Neural Network (RNN). The predicted outcomes clearly define the antenna’s performance and could be used for sensing and prediction purposes.

## 1. Introduction

Due to their remarkable radiation features, including symmetrical patterns with no dispersion, high gain, broadband characteristics, and ease of fabrication, metallic parallel plate waveguide theory-based [1] transverse electromagnetic (TEM) horn antennas have been utilized extensively in a variety of applications, e.g., broadband communication system, satellite tracking, radar, electromagnetic pulse radiator, and radiation measurements [2,3,4,5,6]. A coaxial cable, a microstrip line, or a substrate-integrated waveguide were proposed as general feeding topologies in a horn antenna [7,8,9]. The proposed TEM horn antenna features an exponentially tapered layout to optimize the matching impedance between the microstrip feedline to slot line transition. Because of its large bandwidth, it can not only be used for ultra-wideband (UWB) applications but also in radar-based system measurements and broadband communications. The TEM horn antenna has been designed here in a new way using microstrip to slot line and slot line to exponentially tapered parallel plate waveguide theory for electromagnetic wave radiation.

Single and multilayered dielectric substrate microstrip antennas have been designed where several efforts and approaches have been made to increase the gain, beamwidth, and bandwidth with the help of a stacked radiating parasitic patch [10,11], adding substrate integrated waveguide (SIW) slot array [12], and applying metalized via holes to synthesize the horn walls [13]. Either the gain or the bandwidth of this configuration may be optimized by arranging it in an array or adjusting the feeding structure. It is necessary to improve the gain of the radiating patch element without affecting the bandwidth. TEM antennas owing to the bulky 3D horn sizes and their feeding mechanism are inconvenient; hence, it becomes difficult to set them up in places where there is public gathering and spacing becomes crucial, especially at hospitals and at health care centers for the aged. The radar does not disturb people’s privacy and performs well in low light conditions and has exceptional range resolution and immunity to external interference, unlike camera sensors. In such a situation, compact and lightly managed antennas are required to overcome this situation limited by spaces, antenna size, lighting environment, and weight. In this work, we present a high gain and wideband microstrip feed to a slot line with mounted TEM copper strips at the slot edge on a multilayer FR4 substrate antenna that operates in the 6–15.6 GHz frequency range. The fabricated antenna structure has 88.88% bandwidth and a maximum gain of 9.46 dBi. The high gain, directivity, and broad bandwidth assure the antenna can be used in microwave imaging of the sleeping body position on a bed during various sleeping states using artificial intelligence (AI) which is impacting every human being in every sector, revolutionizing the planet. AI is the primary driver of emerging technologies such as microwave imaging, robots, big data, and IoT, allowing humans to reimagine how we evaluate data, integrate information, and use the results to make better decisions. Deep learning is a type of artificial intelligence that comprises computational models made up of numerous processing layers that learn data attributes through several layers of networks utilizing forward and backward propagation methods. Deep learning includes artificial neural networks (ANNs), convolutional neural networks (CNNs), and recurrent neural networks (RNNs). In these experiments, RNN models are used for microwave image data classification and prediction. Machine learning has enormous potential in the realm of antenna design and antenna behavior prediction. With high accuracy it is found to be useful in various electromagnetic applications such as detecting breast cancer by studying radiation patterns, return loss, phase, and impedance [14,15,16].

The paper is presented as follows: Section 2 presents the design of the proposed antenna and its feeding configuration; Section 3 includes the parametric study of simulated and measurement results and discusses the operation of the antenna with experimental setup and dataset outcomes. Finally, Section 5 states the conclusions of the work with its prospective applications.

## 2. Antenna Design

Figure 1 depicts a 3D image of the antenna elements on a layer of low-cost dielectric substrates that is fed by a T-junction power splitter topology-based microstrip line to slot line and slot line to two parallel copper strips. First of all, the copper coating on both sides of the Fr4 substrate layers is removed and 13 layers of substrate are stacked upon one another so the CNC machine can drill through the layers exponentially. The exponentially drilled substrate layers hold the 10 mm wide copper strip as a radiating patch of the antenna on the drilled face to achieve the structure of an exponential horn antenna. The copper strip is adhesive foil tape with a thickness of 0.08 mm where one of its edges is laid within the ground plane of the bottom substrate layer so that they are electrically connected.

The layers are not glued, for ease of insertion of the copper strip within drilled layers. Because the appropriate length of the screw is not available, Teflon cable is used to bind the substrate layers through two drilled holes so that the substrate layers do not slide upon each other and remain firm. A side view of the antenna with copper strips coupled to a drilled wall of dielectric layers is shown in Figure 2a, while Figure 2b displays the first microstrip feeding layer of a dielectric substrate along with Microstrip T junction feedline, etched slots on the ground plane and the radiating adhesive copper patch strips fixed on the exponentially drilled substrate layers. Similarly, Figure 2c,d show the feed section bottom substrate layer and top view of the proposed antenna.

The microstrip feeding section of the antennas comprises the T junction topology-based microstrip to slot line and slot line to two parallel copper strips line power transitions. The antenna was fabricated on Fr4 substrate (εr = 4.4). The antenna is 26.8 mm × 32.63 mm × 21.6 mm in size. The bottom first dielectric layers of thickness 0.8 mm of the antenna have a 50-ohm microstrip feedline that has a width of “Fw” which is followed by a T-junction base with a width of “Wl2”. The base provides a symmetric T-junction power branch where each branch forms a microstrip to slot line and slot line to two parallel exponential copper strips (E_1_, E_2_) transition networks mounted on the perforated faces of 13 dielectric layers. The guided wavelength (λg) of the frequency band determines the base length of the T-junction (λg/4) feed line, etched slot length, length, and width of the substrate. Table 1 lists the parametric value of the antenna structure with the two exponential copper strip curves E1 and E2 calculated using the following equation:(1)$$E_{1}:y=\;\frac{1}{2}(C_{\mathrm{bw}}(\exp\;(z\frac{1-\ln\left(C_{\mathrm{bw}}\right)}{T_{\mathrm{hs}}})))\left(0\leq z\leq T_{\mathrm{hs}}\right)$$
(2)$$E_{2}:y=\;-\frac{1}{2}(C_{\mathrm{bw}}(\exp\;(z\frac{1-\ln\left(C_{\mathrm{bw}}\right)}{T_{\mathrm{hs}}})))\left(0\leq z\leq T_{\mathrm{hs}}\right)$$

## 3. Result and Discussion

The suggested antenna is optimized and simulated using commercially available high-frequency structure simulator (HFSS) software. Figure 3 depicts the simulation and measurement outcomes. Figure 3a indicates the proposed antenna’s impedance bandwidth of less than 10 dB from the 6 GHz to 15.6 GHz range. The maximum depth at the measured return loss is observed around 10 GHz, which corresponds to the antenna’s resonating frequency at the guided wavelength in a microstrip patch that is the sum of the ground slot radius and the slot length (Cw + Fw + Sr = 13.94 mm). The remaining deeps are created either by the harmonics or resonance in the radiating patch. Figure 3b shows the realized gain, which is within 9.46 dBi over the entire bandwidth and is in close agreement with the measured and simulated results. Except at 8, 10, 12, and 14.5 GHz, the simulated and measured results of the realized gain differ significantly, which can be attributed to connection losses due to dimension imperfections and parasitic effect, improper soldering of the feed line to the connector, oxide layer on the copper surface, parasitic effect existing in between the touched layer of copper foil strips and ground base layer and fabrication faults during the etching process and characterization of the substrate parameters. Figure 3c,d indicate the simulated return loss, realized gain, and efficiency of the bottom substrate feed layer as shown in Figure 2c. The power loss is caused by the impedance mismatch at the feeding section of the base substrate, which is mitigated by placing substrate layers above the ground plane. Placing the exponentially drilled substrate layer with a copper strip within it enhances not only the S-parameters of the antenna but also the gain and directivity. The simulated radiation efficiency of the proposed antenna is within the 82 to 53 percentage.

Figure 4 displays the fabricated antenna’s measured and simulated 2D radiation patterns at 5, 6.5, 7.5, 8, 9.5, 10.5, 11.5, 12.5, 13.5, and 14.5 GHz frequencies. One of the prerequisites for a horn antenna array is that the radiation patterns of the antenna be essentially directive in both the E-plane (*y*-*z* plane) and H-plane (*x*-*z* plane).

The simulated electric field distribution at 6.8 GHz frequency with 3D radiation plots is shown in Figure 5a,d. As can be seen, each tapered copper plate surface of the horn array experienced a shift in surface current, which induced the electric field to radiate. These fields are superimposed to create the plane-like wave that is transmitted in the direction of wave propagation. The radiated field is directive with a unidirectional 3D radiation pattern. To meet microwave imaging applications, the antenna must emit the maximum amount of radiation energy in the desired direction. This is verified by one of the most important measurement results of a high front-to-back ratio at the operating bandwidth of the E-plane beam of the antenna. Figure 5b displays the front-to-back ratio and half-power beamwidth or 3 dB beamwidth of the proposed antenna. The front-to-back ratio of this fabricated antenna is in the range of 3.52 to 23.8 dB within the intended band whereas half-power beamwidth fluctuates within the range from 24.83 to 92.5 degrees. The surface wave propagating along the fixed size of the copper strip and the ground surface plane will radiate and superimpose to produce differing degrees of side-lobe and back-lobe, causing the front-to-back ratio to vary with frequency. The analysis of the phase response of the return loss (S21) is crucial in wideband antenna design, especially for microwave imaging applications. Figure 5e depicts the test setup outcomes for two identical horn antennas in the far field separated by 20 cm to confirm the linear phase response over the operating frequency band. For the measurement, the two antennas are aligned face to face and side by side, and the stable variation of the phase response of S21 is achieved within the working frequency range, signifying minimal dispersion acceptable for radar applications. Figure 5c shows the anechoic chamber measurement setup of the fabricated antenna.

In Table 2, the proposed antenna’s size, radiation performance, and feed system are compared with those of existing antennas. The proposed antenna provides a better antenna, feed management, compact size, and gain in contrast to the current ultra-wideband antenna and its array.

## 4. Experimental Study and Results

After validating the proposed antenna design, the experiment was performed in a room, in a controlled configuration measuring setup for microwave imaging. The objective of the test is to analyze and evaluate the different states of the body on the bed. These experiments are designed to enhance patient monitoring techniques to know the sleeping state of the patient on the bed. The hospitalized elderly patients need to be taken care of to prevent them from falling off the bed and constant monitoring through real-time inspections has to be done which is time-consuming and tedious work. Hence, the goal is to measure and evaluate the change in received signals that get reflected from various sleeping states on the bed. These signals are investigated to build up the four state-of-the-body (while resting on the bed) detection techniques using existing artificial neural networks for classification and prediction.

### 4.1. Experimental Setup

As illustrated in Figure 6a, the setup includes a UWB radar module (NVA-X2 R661 from Xethru Co., Oslo, Norway), a tripod stand, RF cables, and connectors connecting to the PC and proposed antenna modules, and a bed for monitoring various sleeping states. The specimens represent mid-aged human body resting on a hospital-size bed. The bed is 200 cm × 80 cm × 45 cm in size, sectioned by two pieces of paper tape attached 26 cm apart representing the right, middle, and left portions of the bed. At each position, scans were taken: touching the front bar of the bed with the head, touching the back bar with the foot, and in the middle without touching either bar. A human body with a height and width of 170 cm × 55 cm was taken as a target for measuring body state at different positions. A measurement was taken for the sleeping state arranged on the right, middle and left side of the bed and one without a human presence on the bed without shifting the radar module positions. The radar configuration is fixed, and the body position is stable throughout the signal-analyzing process for each stage. The X2 chip of the NVA-R661 module, as shown in Figure 6b, generates and transmits UWB pulses of high-order Gaussian impulse signal with several GHz bandwidths of signal duration in the order of nanoseconds [24]. The high-frequency signal was chosen based on the maximum gain and return loss of the antenna, as well as the radar’s ability to scan with great precision across the bed. The Xethru radar module’s PGSelect command is used to adjust the list of frequencies of the transmitted impulse signal. The impulse signal was set to around the operating bandwidth of the antenna by selecting number 5 of the PGSelect command, which has a center frequency ($f_{c}$) at 6.8 GHz, 2.3 GHz bandwidth with a peak-to-peak output amplitude of 0.69 volts and mean average output power around −12.6 dBm at a pulse repetition frequency of 100 MHZ. This setup met the minimum separation of antenna distance, $d_{c}$ = 4.5 cm residing parallel to each other attached through a SMA connector to the radar module facing towards the sleeping target and fulfilling the antennas separation criterion $d_{c}$ > $\lambda_{c}/4$. The number of frames per sample was set to 200. The radar resolution per frame depth was set to 0.41 cm in air. Figure 7 shows the time and frequency domain response of the transmitted signal pulse in order of nanoseconds for the selected ($f_{c}$) at 6.8 GHz.

### 4.2. Signal Analysis

Background signals and noise are inherent in the transmitted signals as they are reflected through the resting body, bed, and surroundings, so the received signals are significantly attenuated. The received raw signal is cross-correlated with the template signal (Figure 7a), which is generated by the X2 chipset to yield a normalized correlated signal as shown in Figure 8 representing signal strength with and without a sleeping body on the bed. This aids the signal in obtaining the maximum signal-to-noise ratio and correlates the received signal pattern with a low signal variation. This method of extracting the raw signal with a correlated signal has been also used in the detection of hidden targets inside the voids of concrete brick and polar fluid viscous properties measurements [25,26]. The measurement was made using a total of 50 samples of correlated signals and grouped covering the whole specimen as a single bin. Since each body state measurement comprises 207 bins, a total of 10,350 samples were captured for each sleeping body state. A total of 41,400 samples are employed for training, validating, and testing purposes across four distinct body states. The 2-D space image of 50 correlated samples obtained for every sleeping state is plotted in Figure 9. In a total of 207 bins for each sleeping state, 69 bins included touching the head on the front bar of the bed, 69 bins touching the rear bar of the bed, and 69 bins at the middle without touching the front and back bar. The spikes and variations in the image are due to the movement of hands, chest, or feet during the measurement process. Sleeping positions at the right and middle position on a bed-related scanned image have a distinguishable pattern as compared to the remaining state’s position. The left position and without body presence state are visually indistinguishable but can be distinguished by the correlated signal levels. These signal levels on each bin of the samples are the training and testing data sets for RNN models.

### 4.3. Position Results Using Standard Recurrent Neural Network (RNN)

From a total of 828 bin samples of different sleeping body states, 70% of the total randomly chosen bin sets were used as the training set, 15% of the sets as a validation set, and the remaining 15% of the set as a test set. The expected output is predicted sets of four positions for each of the input’s one bin per state (left, middle, right, and empty bed state). MATLAB is used for training and computing the neural results. 25 epochs were used, and the learning rate was set to 0.001. The hyperbolic tangent activation function was used while maintaining the number of nodes in a hidden layer at 8. Figure 10 depicts the validation accuracy and classification loss as the number of iterations increased from 0 to 1800 with 72 iterations per epoch using a Bidirectional long-short-term memory-gated recurrent neural network. There could be multiple optimum solutions for deep networks with significant structural variations, which enables the accuracy and loss curve as seen in Figure 10 to flutter before becoming consistent after some iterations. RNN layers such as Long Short-Term Memory (LSTM) and its extended versions Gated Recurrent Unit (GRU), and Bidirectional Long Short-Term Memory (BiLSTM) are used for training and validations. These layers are feed-forwarding models useful for learning dependencies between time steps in time series or sequential data based on analysis of the previous inputs of the sequence performing superior analysis. Traditional ANNs do not connect the past with the present; however, RNN-based models have different flavors depending on how well they can recall the prior input data of longer sequences.

The Confusion Matrix for the suggested sleeping body states recognition framework is shown in Figure 11. Each body state categorization accuracy is displayed diagonally in green, while the error values are displayed in light red. As we can see, the sleeping state of the body to the right, middle and empty bed positions create significantly less varied states in comparison to the left side, indicating a higher level of accuracy. Compared to the left, middle, and right bed states, the empty bed state position has been more predicted than its true state.

### 4.4. Comparison with RNN Techniques

Table 3 compares the results with those of existing RNN techniques for performance evaluation. Using the obtained radar data, we first evaluated the accuracy of LSTM, GRU, and BiLSTM RNN layers. The validation accuracy of using these three layers for the training set was 97.6%, 98.38%, and 99.19%, respectively. These results demonstrate that BiLSTM algorithms surpass one-way LSTM. Aside from this, GRU appears to be an average model. In predicting states of resting body posture, BiLSTM outperforms standard LSTM and GRU.

## 5. Conclusions

A uniplanar T-junction power divider approach feed system consisting of a metallic parallel plate waveguide theory-based TEM horn-shaped copper strip antenna placed on an exponentially carved multilayer substrate has been presented. The compact antenna is in 3D shape, size, and lightweight and exhibits a T-junction power divider feeding methodology operating over a large frequency range. The feeding has a T-junction formed by a microstrip line to a slot line where the edge of the slot line is mounted by two TEM copper strips exponentially tapered parallel plates on multilayer FR4 substrates. The proposed antennas have maximum realized gain up to 9.46 dBi and a 3 dB beamwidth range from 24.83 to 92.5 degrees, around ultra-wideband regions of bandwidth from 6 to 15.6 GHz. Broad bandwidth, high gain, strong directivity exhibiting the front-to-back ratio between 3.52 dB to 23.81 dB, simple antenna size, and weight are all benefits of the suggested antenna, making it one of the viable options for prospective applications that need to detecting voids and targets in a buried place, for microwave imaging, as well as for detecting heart and breathing rates along with body positions and movements. The microstrip feeding system is compact with the proper position of a two TEM horn antenna, superimposing the radiated beam, and significantly enhancing the radiation directivity of the antenna. The fabricated antenna is implemented in a microwave imaging of the state of the resting body in various sleeping positions.

To begin, radar scans were taken on the resting body on the bed, and the scanned signal for various sleeping state samples per bin was calculated. RNN was used to train and classify body states from scanned radar sample data. The classification result showed that this method can be used for determining different sleeping position states which can be useful for monitoring elderly patients at hospitals and at health care centers for the aged.

## Figures and Tables

**Figure 1 sensors-22-09555-f001:**
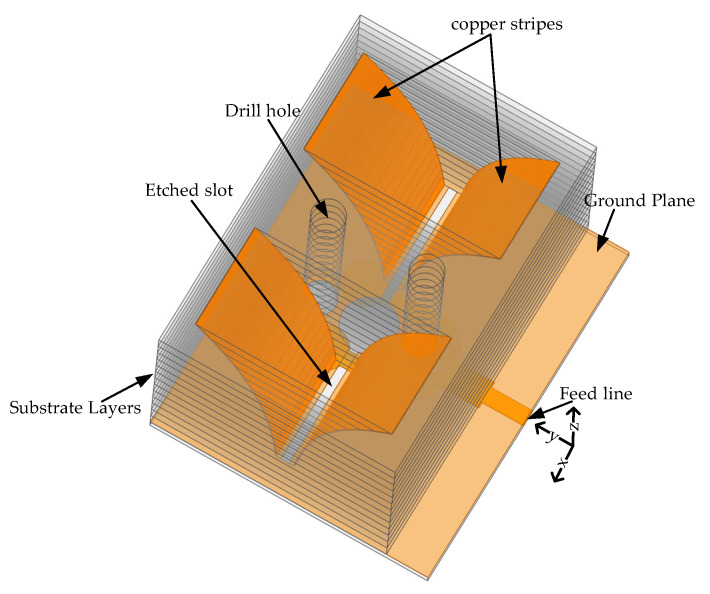
Horn antenna engraved on a multilayered dielectric substrate. 3D perspective view.

**Figure 2 sensors-22-09555-f002:**
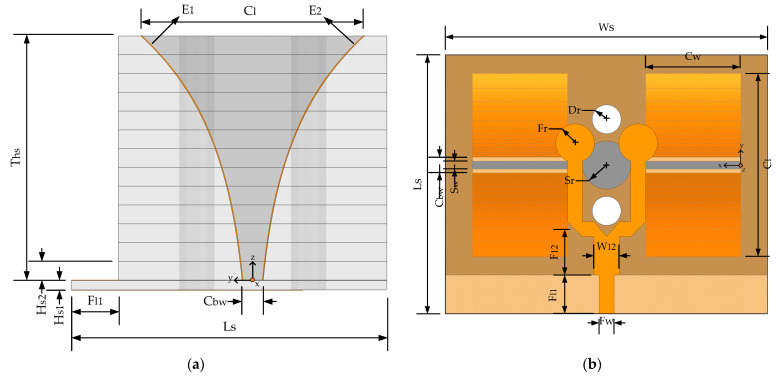
Structure of antenna (**a**) Side view of the antenna with exponentially shaped adhesive foil tape on the dielectric substrate layers; (**b**) Bottom view of the antenna showing feed line, slots, drill holes, and copper strips; (**c**) Front and back view of the bottom substrate layer of thickness 0.8 mm; (**d**) Top view of the fabricated antenna.

**Figure 3 sensors-22-09555-f003:**
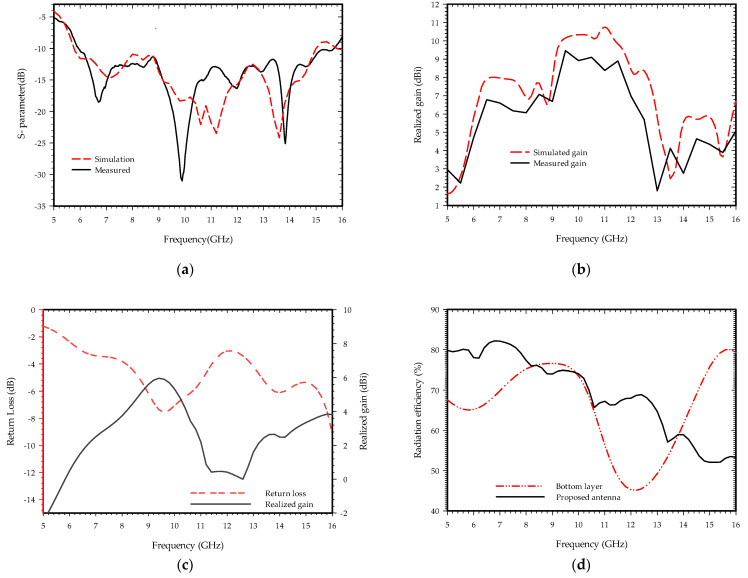
The simulated and measured result of the proposed antenna; (**a**) Return loss; (**b**) Realized gain; (**c**) Simulated return loss and realized gain of the bottom substrate feed layer; (**d**) Radiation efficiency of the proposed antenna and bottom substrate feed layer.

**Figure 4 sensors-22-09555-f004:**
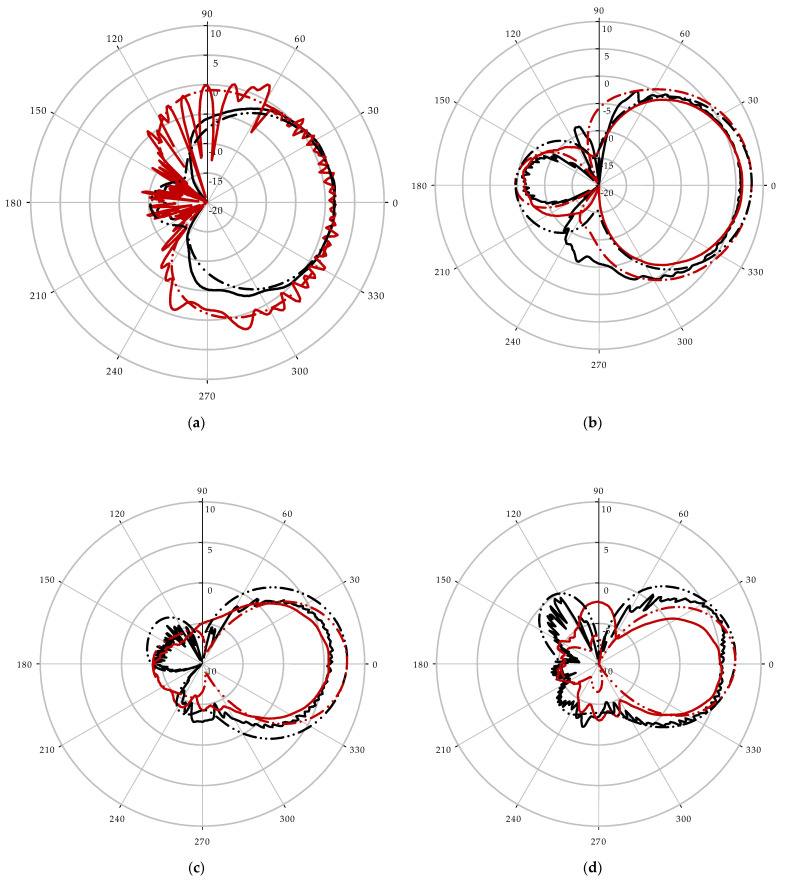
Measured far-field radiation pattern at E-plane and H-plane: (**a**) 5 GHz; (**b**) 6.5 GHz; (**c**) 7.5 GHz; (**d**) 8 GHz; (**e**) 9.5 GHz; (**f**) 10.5 GHz; (**g**)11.5 GHz; (**h**) 12.5 GHz; (**i**) 13.5 GHz; (**j**) 14.5 GHz.

**Figure 5 sensors-22-09555-f005:**
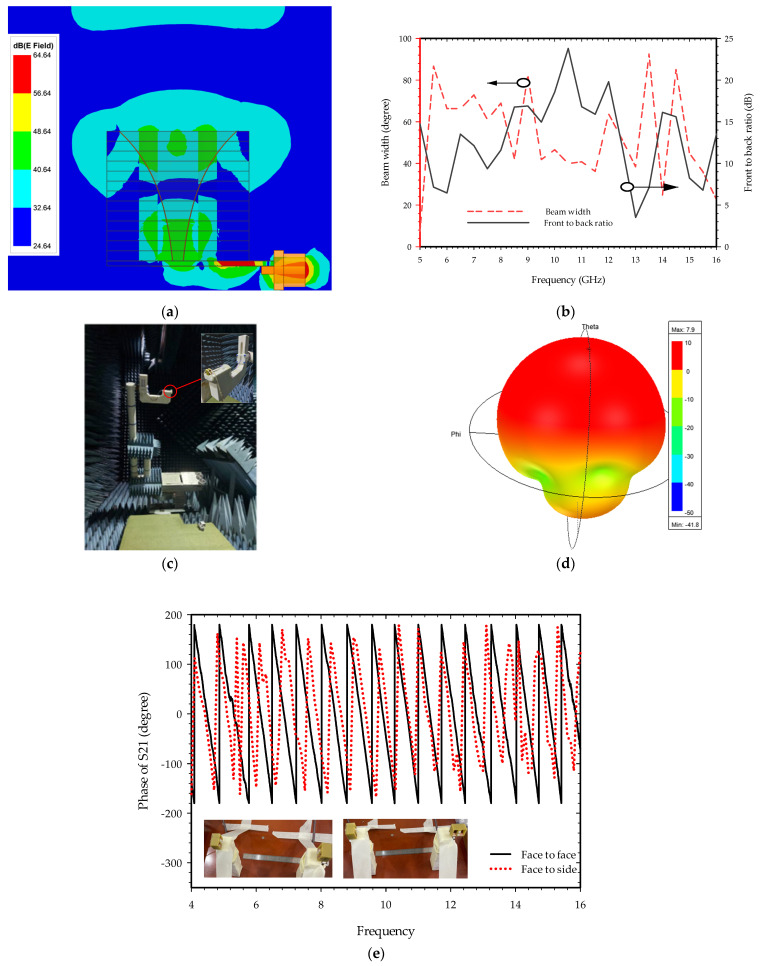
Far-field radiation beam components: (**a**) Electric field distribution plot at 6.8 GHz; (**b**) Front-to-back ratio and beam width; (**c**) Anechoic chamber measurement setup; (**d**) 3D radiation plot at 6.8 GHz; (**e**) Measured phase response of S21.

**Figure 6 sensors-22-09555-f006:**
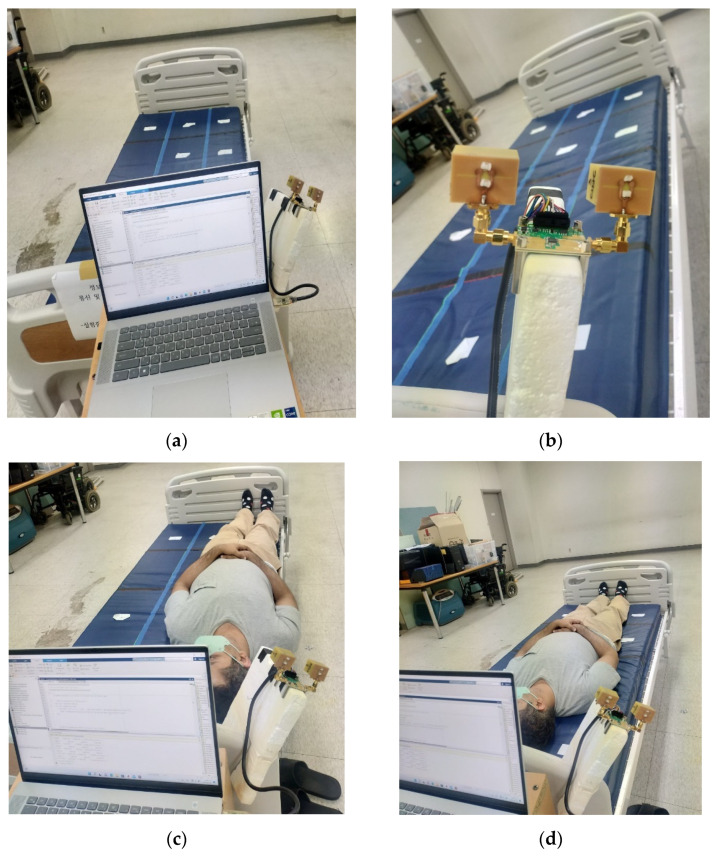
Experimental setup and sleep positions and states: (**a**) Arrangement setup with antenna, Rf cables, PC, with empty bed state; (**b**) UWB radar module with a pair of antennas; (**c**) Right side of the bed; (**d**) Middle of the bed.

**Figure 7 sensors-22-09555-f007:**
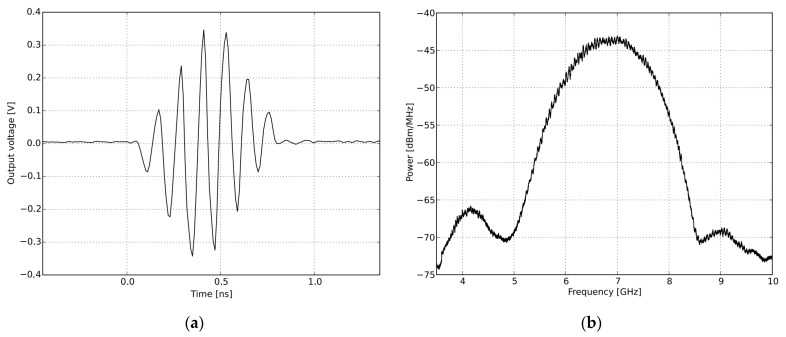
Time and frequency domain pulse shaped of IR-UWB radar for PGselect = 5; (**a**) Transmitted pulse shape in the time domain; (**b**) Transmitted impulse in the frequency domain.

**Figure 8 sensors-22-09555-f008:**
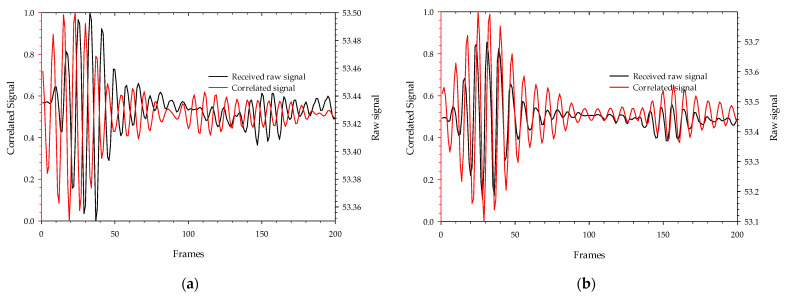
The received pulse signal level with and without correlation obtain from 6.8 GHz transmission: (**a**) Without sleeping body; (**b**) With sleeping body.

**Figure 9 sensors-22-09555-f009:**
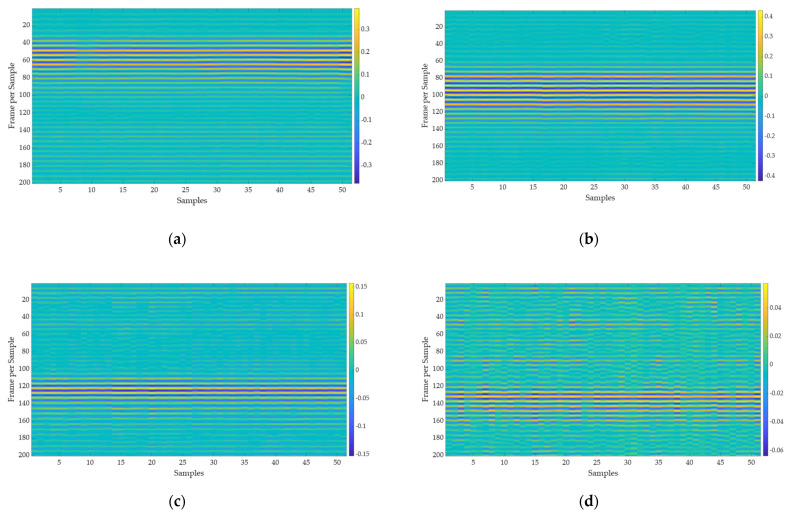
Scanned 2D holographic slices at different sleeping body positions in bed: (**a**) Right position; (**b**) Middle position; (**c**) Left position; (**d**) Without a presence in the bed.

**Figure 10 sensors-22-09555-f010:**
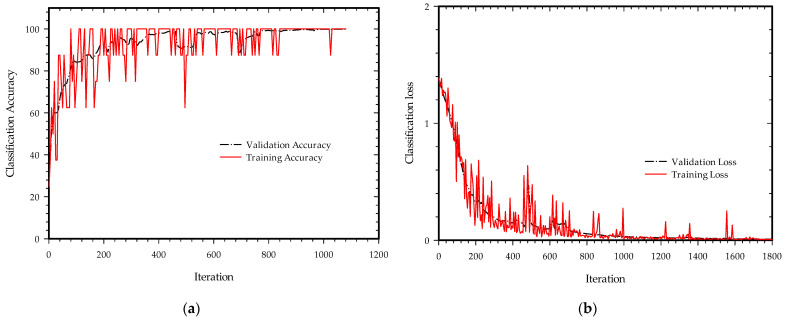
Training progress with 25 epochs: (**a**) Classification accuracy; (**b**) Classification loss.

**Figure 11 sensors-22-09555-f011:**
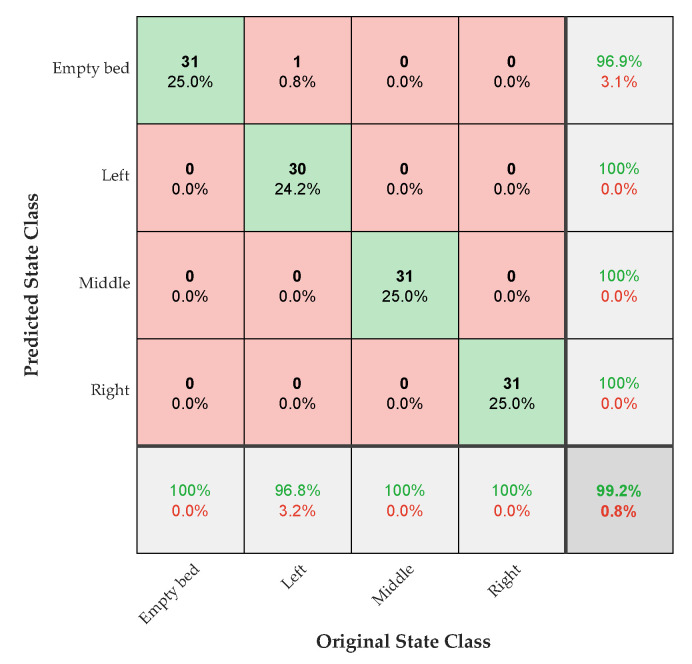
Classification accuracy for each sleeping state, represented by a Confusion Matrix for the Bi-LSTM layer.

**Table 1 sensors-22-09555-t001:** Dimension of the proposed antenna design.

Parameter	mm	Parameter	mm
λg	19	Fl1	4
Ls	6 (λg/5) + Fl1	Fl2	λg/4
Ws	2 (Sr + Fw + Cw) + λg/4	Wl2	2.63
Hs1	0.8	Fr	2
Hs2	1.6	Sr	2.5
T_hs_	13 (Hs2)	Dr	1.5
Fw	1.54	Cbw	Sw + 1
Cl	19	Sw	0.8
Cw	(λg + Sw)/2	---	---

**Table 2 sensors-22-09555-t002:** Comparison table based on antenna bandwidth, feed system, size, and gain.

Ref.	Frequency Range (GHz)	Feed System	Size (mm^2^)	Gain (dBi)
[17]	2–12	A microstrip-to-parallel strip balun	63.2 × 60.4 × 80	--
[18]	1.54–5.29	Co-axial cable	61.6 × 61.6 × 108	7.8
[19]	2–14	Co-axial cable	74 × 74 × 80	11.52
[20]	6–18	Adaptor based on double-ridged waveguide	29.58 × 22.97 × 36.1	10.75
[21]	12.25–12.75 and 14.0–14.5	Substrate-integrated coaxial line	26 × 26 × 26.3	7.7 and 9.2
[22]	4–6	1 × 8 tapered arc strip impedance transformer lines	54.4 × 54.4 × 85.7	5.7
[23]	1.45–5.78	4 × 4 butler matrix	70.65 × 90 × 140	9.18
Proposed antenna	5.6–15.6	T-junction power divider with slot to radiating plane transition	26.8 × 32.63 × 21.6	9.46

**Table 3 sensors-22-09555-t003:** Accuracy comparison with RNN Layer.

RNN Layers	Validation Accuracy
LSTM	97.6
GRU	98.38
BiLSTM	99.19

## Data Availability

The data presented in this study are available on request from the corresponding author.

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
