# Peer review of "Compact Wideband Double-Slot Microstrip Feed Engraved TEM Horn Strip Antennas on a Multilayer Substrate Board for in Bed Resting Body Positions Determination Based on Artificial Intelligence"

_sensors, 2022, doi:10.3390/s22239555_

Round 1
Reviewer 1 Report
In this paper, a horn-shaped stripe antenna exponentially tapered carved on a multilayer 12 dielectric substrate for an indoor body position tracking system is proposed. Please check the following comments.
1) The authors should insert the simulated radiation pattern.
2) The advantages of the suggested antenna should be written as a comparison with another type of antenna.
3) The subscription should be fixed.
Author Response
The reviewer's valuable comments are reviewed and incorporated into the paper.
Reviewer 2 Report
The submitted paper introduces a novel horn-shaped metallic stripe antenna exponentially tapered on a multilayer dielectric substrate for an indoor body position tracking system through artificial intelligence. The paper has been structured well. I would ask the authors to check the syntax errors in the manuscript. It would be well to include the phase of S21 which has to be measured for two identical antennas in the far field in order to confirm the linear phase behaviour over the operation frequency band. Otherwise the same imaging performance could not be obtained for the different antenna locations with the varying separation distances in reference to the body positioning object. Can you please correct the typing error in Figure 8a?
Author Response
The reviewer's valuable comments are reviewed and incorporated into the paper.
The errors have been identified and corrected and phase response S21 has been added with correction in figure 8(a).
Reviewer 3 Report
All remarks are included in attached file.

Reviewer 4 Report
The authors of this paper reported a Compact wideband Double-Slot microstrip feed engraved TEM horn stripe antennas on a multilayer substrate board for in bed resting body positions determination based on artificial intelligence. This work was done with good effort, however, there are minor issues that need to be addressed. After responding to the comments, I recommend the paper be accepted.
1- The organization of the paper should be revisited, for example the figures should be below the corresponding discussion paragraph.
2- Regarding equations 1 and 2, did you derive them or taken from a reference?
3- Figure 5.c is blurred. It is better to have a clearer one.
Author Response
Thanks for the reviewer's valuable suggestions and comments
1- The organization of the paper should be revisited, for example, the figures should be below the corresponding discussion paragraph
The paper is formatted according to reviewer comments best as possible according to figures content and space occupied.
2- Regarding equations 1 and 2, did you derive them or taken from a reference?
The equations are derived on the basis of exponential equations on the basis of a side view of the antenna (Figure 1a)
3- Figure 5.c is blurred. It is better to have a clearer one.
figure 5 is updated
Round 2
Reviewer 1 Report
Now, the revised manuscript is proper this journal
Reviewer 3 Report
Paper is significantly improved, however there are still mistakes at chapters enumeration and also “Bidirectional long-short-term memory (BiLSTM)” dash mistake on line 272 is not repaired.